# Streaming Bayes GFlowNets

**Tiago da Silva**
Getulio Vargas Foundation
tiago.henrique@fgv.br

**Daniel Augusto de Souza**
University College London
daniel.souza.21@ucl.ac.uk

**Diego Mesquita**
Getulio Vargas Foundation
diego.mesquita@fgv.br

## Abstract

Bayes' rule naturally allows for inference refinement in a streaming fashion, without the need to recompute posteriors from scratch whenever new data arrives. In principle, Bayesian streaming is straightforward: we update our prior with the available data and use the resulting posterior as a prior when processing the next data chunk. In practice, however, this recipe entails i) approximating an intractable posterior at each time step; and ii) encapsulating results appropriately to allow for posterior propagation. For continuous state spaces, variational inference (VI) is particularly convenient due to its scalability and the tractability of variational posteriors. For discrete state spaces, however, state-of-the-art VI results in analytically intractable approximations that are ill-suited for streaming settings. To enable streaming Bayesian inference over discrete parameter spaces, we propose streaming Bayes GFlowNets (abbreviated as SB-GFlowNets) by leveraging the recently proposed GFlowNets — a powerful class of amortized samplers for discrete compositional objects. Notably, SB-GFlowNet approximates the initial posterior using a standard GFlowNet and subsequently updates it using a tailored procedure that requires only the newly observed data. Our case studies in linear preference learning and phylogenetic inference showcase the effectiveness of SB-GFlowNets in sampling from an unnormalized posterior in a streaming setting. As expected, we also observe that SB-GFlowNets is significantly faster than repeatedly training a GFlowNet from scratch to sample from the full posterior.

## 1 Introduction

One of the foundations of the Big Data revolution in the sciences and engineering is the use of streaming data that is continuously collected and meant to be processed as it arrives. Many of the large statistical models in use were first formulated for the batched i.i.d. data setting and updating such models in this setting is a challenge of its own, as it would naively require us to revisit all past data at each update to avoid forgetting. [19, 39] In this context, Bayesian inference appears as a natural starting point to learning in the streaming data setting due to its innate property of *coherence* [5], which allows learning can happen continuously, independently of how the data is chunked into packages or ordered. More specifically, given a prior distribution $p(\theta)$ and a data generating likelihood $p(y_i \mid \theta)$, the posterior distribution over a set of data $p(\theta \mid y_1, y_2) \propto p(y_2 \mid \theta)p(y_1 \mid \theta)p(\theta)$, can be written as $p(y_2 \mid \theta)p(\theta \mid y_1)$, where first we compute the posterior over $y_1$ and then used as a prior to compute the posterior over $y_2$. As conveniently summarized by Lindley [33]: "Today's posterior is tomorrow's prior".

Nonetheless, Bayesian inference is notoriously known for being hard to obtain closed-form solutions. The gold standard of approximate methods, Markov chain Monte Carlo (MCMC), requires that the prior and likelihood distributions probability density/mass functions can be evaluated, however,

as it can only obtain samples from posterior, using the previous posterior as the prior of the next time-step requires additional work [6]. Thus, variational inference (VI), another popular approximate Bayes method, appears as the best fit as its foundation lies in directly parameterizing the approximate posterior distribution instead of relying on empirical distributions of samples.

Importantly, many problems of interest are built upon a discrete set of parameters. In Bayesian phylogenetic inference (BPI), for example, we are concerned with topologies of phylogenetic trees describing the evolutionary genetic history of a population. This is a especially compelling application of streaming Bayes, allowing researchers to update posteriors over the phylogenetic trees as they decode new nucleobases in genetic sequences — without reprocessing previously decoded nucleobases. In practice, phylogenetic analyses might involve hundreds of thousands of nucleobases.

Nonetheless, popular methods for VI over discrete posteriors typically rely on gradient-based optimization (e.g., to maximize evidence lower bounds), requiring the modeling of discrete parameters as transformed versions of latent continuous ones [20, 25, 36]. For methods using VI, a discrete posterior distribution with support over a set of size $N$ would require variational posteriors of dimension $N$, implying either the storage of large covariance matrices of order $N^2$, in the case of continuous reparameterizations, or the use of massively simplifying assumptions such as mean-field, reducing the expressivity of the variational posterior. As an example of support size for discrete distributions, the number of graphs with $N$ nodes grows asymptotically as $\sqrt{2^{N(N-1)}}/N!$ [16].

In this work, we leverage Generative Flow Networks (GFlowNets) [2, 3, 30] to efficiently address the implementation of approximate Bayesian models defined on a discrete set of parameters within a streaming data setting. In summary, GFlowNets are a family of amortized VI methods for high-dimensional discrete parameter spaces, that learn the policies of a finite-horizon Markov decision process (MDP) by minimizing a loss function enforcing a *balance condition*. The condition, in turn, provably ensures that the samples generated from the MDP are correctly distributed. Notably, despite their successful deployment in solving a wide range of problems ([11, 12, 22–24, 32, 34, 64]), previous approaches assumed that the target (posterior) distribution did not change in time. Hence, this is the first work handling the training of GFlowNets in dynamic environments.

More specifically, we propose two novel training schemes to enable GFlowNets on streaming settings. The first consists of directly enforcing a *streaming balance* condition, which induces a least-squares loss that can be optimized in an off-policy fashion, potentially avoiding issues such as mode collapse. Alternatively, we leverage the well-known relationship between GFlowNets and VI to develop an on-policy divergence-minimizing algorithm that often exhibits faster training convergence when the target distribution is not very sparse. We also analyze theoretically how local errors accumulate through posterior propagation, providing upper bounds on the approximation quality of models learned using both of our update schemes.

In summary, our main contributions are:

1. We propose a streaming balance condition and a corresponding provably correct algorithm enabling the training of GFlowNets in a streaming data setting (Algorithm 1), without the need to revisit past data;

2. We devise an alternative VI algorithm employing low-variance gradient estimators to train to update GFlowNets in streaming fashion (Section 3). ;

3. We theoretically analyze how inaccuracies of individual updates propagate to subsequent posterior approximations, bounding the approximation errors accumulated through repeated streaming updates of the model (Section 4);

4. We demonstrate the correctness and effectiveness of our method in a series of streaming tasks, such as Bayesian linear preference learning with integer-valued features (Section 5.2) and online Bayesian phylogenetic inference (Section 5.3).

## 2   Preliminaries

**Notation and definitions.** Let $\mathcal{X}$ be a set of compositional objects (e.g., trees with nine nodes) and $\mathcal{S} \supseteq \mathcal{X}$ be an extension of $\mathcal{X}$ (e.g., forests with nine nodes). Define $\mathcal{G} = (\mathcal{S}, \mathbf{A})$ as a directed acyclic graph (DAG) with nodes in $\mathcal{S}$, adjacency matrix $\mathbf{A}$, and the following properties: (i) $\mathcal{G}$ is weakly connected; (ii) there is an unique state $s_o \in \mathcal{S} \setminus \mathcal{X}$ without any incoming edges; and (iii) there are no edges leaving $x$ for every $x \in \mathcal{X}$. We call $\mathcal{S}$ the *set of states*, $\mathcal{G}$ the *state graph*, and

$\mathcal{X}$ the *set of terminal states*; $s_o$ is called the *initial state*. A *forward policy* over $\mathcal{G}$ is a function $p_F(v, w)\colon \mathcal{S} \times \mathcal{S} \to \mathbb{R}_+$ such that it defines a probability distribution over $\mathcal{S}$ with support on $v$'s children in $\mathcal{G}$. A *backward policy* is a forward policy over $\mathcal{G}^\intercal = (\mathcal{S}, \mathbf{A}^\intercal)$. To alleviate notation, we denote the set of trajectories that ends in $x$ as $\{\tau \mid \tau \rightsquigarrow x\}$, and define a probability distribution in this set as $p_F(\tau) = \prod_i p_F(s_i, s_{i+1})$ and the probability of the backwards trajectory condition on the end point $x$ as $p_B(\tau \mid x) = \prod_i p_B(s_{i+1}, s_i)$.

**GFlowNets.** We define a GFlowNet as a tuple $G = (\mathcal{G}, p_F, p_B, Z)$ specifying a state graph $\mathcal{G}$, forward $p_F$ and backward $p_B$ policies, and an estimate $Z$ of a normalizing constant. A GFlowNet induces a probability distribution over $\mathcal{X}$ through its forward policy, $p_\top(x) \coloneqq \sum_{\tau \rightsquigarrow x} p_F(\tau)$. When there is no risk of ambiguity, we will generally omit $\mathcal{G}$ from this notation. We also drop $Z$ when training the GFlowNet with a criterion that does not require it. Following the common convention, we parameterize the forward network by a neural network, and the backward policy at each state is left fixed as a uniform distribution.

The task a GFlowNet is trained for is to match its induced distribution $p_\top(x)$ with a target distribution $\pi(x) \propto \tilde{\pi}(x)$, where $\tilde{\pi}(x)$ is an unnormalized distribution. In practice, the agreement between $p_\top$ and $\pi$ can be enforced by a balance condition over trajectories, avoiding references to $p_\top$. For instance, we may enforce *trajectory balance* (TB) condition, $Z p_F(\tau) = p_B(\tau \mid x)\tilde{\pi}(x)$ for all $\tau \rightsquigarrow x$, to estimate the parameters of the GFlowNet by minimizing

$$\mathcal{L}_{TB}(G) = \mathbb{E}_{\tau \sim \xi}\left[\left(\log \frac{Z p_F(\tau)}{p_B(\tau \mid x)\tilde{\pi}(x)}\right)^2\right], \tag{1}$$

where $\xi$ is a base policy with full support on the space of complete trajectories, i.e., trajectories starting at $s_o$ and terminating at $s_f$. Then, we can use the following unbiased estimate to compute the GFlowNet's marginal distribution over terminal states $p_\top$:

$$p_\top(x) = \mathbb{E}_{\tau \sim p_B(\cdot|x)}\left[\frac{p_F(\tau)}{p_B(\tau|x)}\right] \approx \frac{1}{K}\sum_{1 \le i \le k} \frac{p_F(\tau_i)}{p_B(\tau_i|x)}. \tag{2}$$

For a thorough review of GFlowNets, please refer to [3].

**GFlowNets and VI.** Malkin et al. [38] showed that the training of GFlowNets may be framed as a variational inference on the target $p_B(\tau) \propto \tilde{\pi}(x)p_B(\tau|x)$ with $p_F$ as the proposal distribution. Through the data processing inequality, they showed that the minimization of the Kullback-Leibler (KL) divergence between trajectory-level distributions $p_F$ and $p_B$ incurred in a marginal $p_\top$ over $\mathcal{X}$ matching the target, namely, $\mathcal{D}_{KL}[p_F \| p_B] \ge \mathcal{D}_{KL}[p_\top \| \pi]$. Then, they demonstrated the equivalence between the gradients of the on-policy TB loss and the KL divergence, and that the latter is a sound learning objective for GFlowNets. Here, we build upon this construction to design a KL divergence-based algorithm for updating GFlowNets in a streaming context.

**Problem description.** We assume that the data is drawn from a distribution $f(\cdot|x)$ indexed by a parameter $x \in \mathcal{X}$ and we define $\pi(x)$ as a prior distribution over $\mathcal{X}$. Let $(\mathcal{D}_i)_{i \ge 1}$ be a sequence of continually collected independent data sets and $\pi_t(x|\mathcal{D}_1, \dots, \mathcal{D}_t) \propto f(\mathcal{D}_1, \dots, \mathcal{D}_t|x)\pi(x) \coloneqq \tilde{\pi}_t(x)$ be the posterior conditioned on the first $t$ data sets. Note that, due to coherence, we can also write $\tilde{\pi}_t(x) = f(\mathcal{D}_t|x)\tilde{\pi}_{t-1}(x)$.

## 3 Streaming Bayes GFlowNets

In this section, we propose *streaming Bayes GFlowNets* (SB-GFlowNets) as an extension of GFlowNets to handle streaming data. In essence, an SB-GFlowNet $G_{t+1}$ avoids evaluating $\tilde{\pi}_{t+1}(x)$, which would require revisiting all the previous data, by maintaining an SB-GFlowNet $G_t$ which targets previous posterior $\tilde{\pi}_t(x)$, allowing the target distribution of $G_{t+1}$ to be $f(\mathcal{D}_{t+1}|x)p_\top^{(t)}(x)$, where $p_\top^{(t)}(x)$ is the induced distribution of $G_t$, thus avoiding the storage of previous data sets.

We propose two strategies for matching the target distribution, enforcing the *streaming balance (SB) condition* (Section 3.1) and applying *KL streaming updates* (Section 3.2). Importantly, both methods require only newly observed data $\mathcal{D}_{t+1}$ and the previously trained model, $G_t$ to update the posterior approximation, without revisiting past data batches $\mathcal{D}_{1:t}$. These conditions are extensions of TB condition and KL criterion proposed for the batched data case, each with its own strengths and weaknesses. Subsequently, we provide both a theoretical analysis (Section 4) and experimental validation (Section 5) for SB-GFlowNets.

## 3.1 Streaming balance condition.

In streaming settings, training a GFlowNet to sample from $\tilde{\pi}_{t+1} \propto f(D_t, \ldots, D_1|x)\pi(x)$ by naively enforcing the TB condition entails numerous evaluations of the complete likelihood. However, assuming we have previously trained a GFlowNet to sample from $\tilde{\pi}_t(x)$, we propose a more convenient balance condition that does not refer explicitly to previously-seen data.

**Definition 1** (Streaming balance condition). Let $G_t = (\mathcal{G}, p_F^{(t)}, p_B^{(t)}, Z_t)$ be a GFlowNet trained to sample proportionally to the posterior $\tilde{\pi}_t(x)$. The *streaming balance (SB) condition* for the GFlowNet $G_{t+1} = (\mathcal{G}, p_F^{(t+1)}, p_B^{(t+1)}, Z_{t+1})$, conditioned on $G_t$, is defined as

$$Z_{t+1}p_F^{(t+1)}(\tau) = \frac{f(\mathcal{D}_{t+1}|x)p_F^{(t)}(\tau)Z_t}{p_B^{(t)}(\tau|x)}p_B^{(t+1)}(\tau|x), \tag{3}$$

in which $f(\mathcal{D}_{t+1}|x)$ is the model's likelihood. When the backward policy does not depend on $t$, e.g., is fixed as an uniform policy, Equation 3 reduces to $Z_{t+1}p_F^{(t+1)}(\tau) = Z_t p_F^{(t)}(\tau)f(\mathcal{D}_{t+1}|x)$.

Intuitively, if we consider the TB conditions for the GFlowNets $G_t$ and $G_{t+1}$:

$$Z_t p_F^{(t)}(\tau) = \tilde{\pi}_t(x)p_B^{(t)}(\tau|x) \quad \text{and} \quad Z_{t+1}p_F^{(t+1)}(\tau) = f(\mathcal{D}_{t+1}|x)\tilde{\pi}_t(x)p_B^{(t+1)}(\tau|x),$$

we can re-arrange the identity for $G_t$ to obtain the unnormalized posterior a time $t$, i.e., $\tilde{\pi}_t = {Z_t p_F^{(t)}}/{p_B^{(t)}}$ and, then, apply this to the TB condition of $G_{t+1}$ yielding Equation 3.

Naturally, the SB condition gives rise to a loss function by considering the log-squared ratio between the left- and right-hand sides of Equation 3, namely:

$$\mathcal{L}_{SB}(G_{t+1}; G_t) = \mathop{\mathbb{E}}_{\tau \sim \xi}\left[\left(\log \frac{Z_{t+1}p_F^{(t+1)}(\tau)}{p_B^{(t+1)}(\tau|x)} \cdot \frac{p_B^{(t)}(\tau|x)}{Z_t p_F^{(t)}(\tau)} \cdot \frac{1}{f(\mathcal{D}_{t+1}|x)}\right)^2\right], \tag{4}$$

for a distribution $\xi$ of full-support over trajectories. Importantly, Proposition 1 ensures that this approach results in a model sampling proportionally to $\tilde{\pi}_{t+1}(x)$. In a concurrent work, Venkatraman et al. [54, Eq. (8)] proposed a similar learning objective, termed *relative trajectory balance*, to update diffusion models according to a novel and arbitrary likelihood function.

**Proposition 1** (Soundness of $\mathcal{L}_{SB}$). *Assume $p_\top^{(t)}(x) \propto \tilde{\pi}_t(x)$. Then, if $\mathcal{L}_{SB}(G_{t+1}; G_t) = 0$, then $p_\top^{(t+1)}(x)$ samples objects from $\mathcal{X}$ proportionally to $\tilde{\pi}_{t+1}(x)$.*

Algorithm 1 describes the training of a SB-GFlowNet by minimizing $\mathcal{L}_{SB}$.

## 3.2 Divergence-based updates of SB-GFlowNets.

In practice, estimating $\log Z_t$ by learning is not straight-forward and, if not done properly, may severely damage the quality of the approximation. Divergence-based objectives provide ways to train GFlowNets without relying on estimates of $\log Z_t$. Indeed, as we show in our experimental results (Section 5), minimizing a divergence-based objective often leads to a better approximation than conventional approaches. However, unlike TB which allows the use of arbitrary base policies, these approaches normally require the use of $p_\top(x)$ as the base policy which may lead to mode collapse depending on target distributions [6, 38].

So, as discussed in Section 2, recall that a GFlowNet may equivalently be interpreted as a hierarchical variational model using $p_F(\tau)$ as an approximation to $p_B(\tau) \propto \tilde{\pi}(x)p_B(\tau|x)$ allowing GFlowNets to be trained by minimizing any divergence between $p_F$ and $p_B$ [38]. Based on this insight, we propose the following divergence-based training criterion for streaming GFlowNets:

**Definition 2** (Divergence-based streaming update). Let $G_t = (\mathcal{G}, p_F^{(t)}, p_B^{(t)})$ be a GFlowNet balanced sampling proportional to $\tilde{\pi}_t$. For $G_{t+1} = (\mathcal{G}, p_F^{(t+1)}, p_B^{(t+1)})$ and target distribution $\pi_{t+1}$, define the unnormalized distribution $p(\tau) \propto p_F^{(t)}(\tau)f(\mathcal{D}_{t+1}|x)$ over trajectories. Then, if $\overset{C}{=}$ denotes equality up to an additive constant,

$$\mathcal{L}_{KL}(G_{t+1}; G_t) = \mathop{\mathbb{E}}_{\tau \sim p_F^{(t+1)}}\left[\log \frac{p_F^{(t+1)}(\tau)}{p_F^{(t)}(\tau)f(\mathcal{D}_{t+1}|x)}\right] \overset{C}{=} \mathcal{D}_{KL}\left[p_F^{(t+1)}(\tau)||p(\tau)\right], \tag{5}$$

is called the *KL's streaming criterion*.

Note that when $\mathcal{L}_{KL}$ vanishes, $p_F^{(t+1)}(\tau) \propto p_F^{(t)}(\tau) f(\mathcal{D}_{t+1}|x)$ for all $\tau \rightsquigarrow x$ and $x \in \mathcal{X}$. Then,

$$p_\top^{(t+1)}(x) = \sum_{\tau \to x} p_F^{(t+1)}(\tau) \propto p_\top^{(t)}(x) f(\mathcal{D}_t|x).$$

Consequently, minimizing $\mathcal{L}_{KL}$ is a sound objective for learning to sample from $\pi_{t+1}$ when $p_\top^{(t)}(x) = \pi_t(x)$. In Section 4, we quantitatively relate the accuracy of $p_\top^{(t+1)}$ with that of $p_\top^{(t)}$ when using either Equation (4) or Equation (5) as learning objectives for the SB-GFlowNet.

**Low-variance gradient estimators for $\mathcal{D}_{KL}$.** Score matching estimates of the divergence's gradients, $\nabla_\theta \mathcal{D}_{KL}$, are of high variance, negatively affecting the convergence speed of the trained model [42, 46, 58]. To circumvent this issue, we rely upon the REINFORCE leave-one-out (RLOO) gradient estimator [41], which employs a sample-dependent control variate to significantly reduce the noiseness of the estimated gradients. More specifically, define $\gamma(\tau) = \log \frac{p_F^{(t+1)}(\tau)}{p_F^{(t)}(\tau) f(\mathcal{D}_{t+1}|x)}$ and let $\tau_1, \ldots, \tau_k$ be independently sampled trajectories from $p_F^{(t+1)}$. Also, denote by $\theta$ the parameters of $p_F^{(t+1)}$. In this context, the RLOO estimator for the gradient of KL's streaming criterion is

$$\frac{1}{k} \sum_{1 \leq i \leq k} \nabla_\theta \gamma(\tau_i) + \frac{1}{k} \sum_{1 \leq i \leq k} \left( \gamma(\tau_i) - \frac{1}{k-1} \sum_{1 \leq j \leq k, j \neq i} \gamma(\tau_j) \right) \nabla_\theta \log p_F^{(t+1)}(\tau), \quad (6)$$

which is an unbiased estimate for $\nabla_\theta \mathbb{E}_{\tau \sim p_F^{t+1}}[\gamma(\tau)]$. Importantly, RLOO is straightforwardly represented as a vector-Jacobian product. Thus, it can be swiftly computed in standard reverse-mode autodifferentiation packages [47], adding a negligible computational overhead to the algorithm.

## 4 Theoretical analysis

While posterior propagation is computationally convenient, preventing training from scratch repeatedly, we should also expect errors to propagate through updates. To better understand the behavior of SB-GFlowNets, we analyze how choosing sub-optimal SB-GFlowNet at time $t$ influences our approximation's quality at time $t + 1$. We quantify goodness-of-fit of SB-GFlowNets' sampling distribution wrt target both in terms of TV and of their expected distance in log space. Since the SB loss and KL updates incur different parameterizations, we analyze separately the cases in which SB-GFlowNets are trained using each loss.

Overall, we establish that the accuracy of a SB-GFlowNet's sampling distribution $p_\top^{(t+1)}$ at time $t + 1$ depends on how close the new forward policy $p_F^{(t+1)}$ is from being optimal, on the size of the new data chunk $\mathcal{D}_{t+1}$, and on the goodness-of-fit of the previous estimate $p_\top^{(t)}$. As expected, our analysis suggests that the negative effects of poorly learned $p_\top^{(t)}$ are negligible when the size of a data chunk is relatively large. Also, when the previous SB-GFlowNet $G_t$ is a poor approximation to the true posterior $\pi_t$, we discuss the benefits of using an earlier and potentially more accurate checkpoint $G_s$ with $s < t$ as a reference for training $G_{t+1}$.

### 4.1 Analysis for SB loss-based training

We first analyze how errors propagate when training SB-GFlowNets by directly enforcing the SB condition, i.e., minimizing Equation 4. To this effect, recall using the SB loss implies learning an estimate of the partition function $Z_t$ as well as the forward policy $p_F^{(t)}$ at each time step $t$. In particular, Proposition 2 measures the consequences of an inaccurately learned $p_F^{(t)}$ and of insufficiently minimizing the SB loss on the $(t+1)$th approximation.

**Proposition 2.** *Let* $G_t = (\mathcal{G}, p_F^{(t)}, p_B^{(t)}, Z_t)$ *and* $G_{t+1} = (\mathcal{G}, p_F^{(t+1)}, p_B^{(t+1)}, Z_{t+1})$ *be pair of GFlowNets. Let also* $\pi_{t+1}(x) \propto \pi_t(x) f(\mathcal{D}_{t+1}|x)$ *and recall that* $\pi \colon \mathcal{X} \to \mathbb{R}_+$ *is our prior distribution. Defining* $Z_k^\star := \sum_x \pi(x) \prod_{i=1}^k f(\mathcal{D}_i|x)$ *for any* $k > 0$*, and letting* $(\hat{p}_F^{(t+1)}, \hat{p}_B^{(t+1)}, \hat{Z}_{t+1})$ *be the optimal solution to Equation 4 satisfying the SB condition. Then,*

$$\delta_{LS}^{\pi_{t+1}} \left( p_\top^{(t+1)}, \pi_{t+1} \right) \leq \underbrace{\delta_{LS}^{\pi_{t+1}} \left( p_\top^{(t+1)}, \hat{p}_\top^{t+1} \right) + \left| \log \frac{\hat{Z}_{t+1}}{Z_{t+1}^\star} \right|}_{\text{Estimation error}} + \underbrace{\delta_{LS}^{\pi_{t+1}} \left( p_\top^{(t)}, \pi_t \right) + \left| \log \frac{Z_t}{Z_t^\star} \right|}_{\text{Accuracy of } p_\top^{(t)}},$$

*where $\delta_{LS}^{\xi}(p,q) := \left(\mathbb{E}_{x \sim \xi}[\log p(x) - \log q(x)]^2\right)^{1/2}$.*

As expected, Proposition 2 underlines the importance of starting with a good approximation from the $t$th stage and of properly solving the learning problem at the $(t+1)$th stage in order to obtain an accurate approximation to $\pi_{t+1}$. In particular, this result reveals the detrimental effect of an inadequately estimated partition function on the quality of GFlowNet's streaming updates.

Complementary, Proposition 3 provides an upper bound the TV distance between $p_F^{(t+1)}$ and $p_B^{(t+1)}(\tau) := p_B^{(t+1)}(\tau|x)\pi_{t+1}(x)$. Besides corroborating the conclusions of Proposition 2, the explicit dependence of the TV upper bound on of $\mathcal{D}_{t+1}$ allows for analyzing it as a function of the newly observed data set's likelihood $f(\mathcal{D}_{t+1}|x)$. In particular, consider the case in which $|\mathcal{D}_{t+1}| \to \infty$. Then, since $f(\mathcal{D}_{t+1}|x) \to 0$ uniformly on $x$ and, for many models, $f(\mathcal{D}_{t+1}|\hat{x})/f(\mathcal{D}_{t+1}|y) \to \infty$ for $y \notin \arg\max_x f(\mathcal{D}|x)$. Then, the accuracy of $p_\top^{(t)}$ becomes negligible to the overall approximation of the full posterior when the newly observed data set $\mathcal{D}_{t+1}$ is relatively large and the second term in Proposition 3 approximately vanishes.

**Proposition 3.** *Let $TV(p,q) := \frac{1}{2}\sum_{x \in \mathcal{X}}|p(x) - q(x)|$ be the TV distance between probability distributions $p$ and $q$. Then, under the same setting of Proposition 2,*

$$TV\left(p_\top^{(t+1)}, \pi_{t+1}\right) \leq TV\left(p_\top^{(t+1)}, \hat{p}_\top^{(t+1)}\right) + \frac{1}{2} \cdot f(\mathcal{D}_{t+1}|\hat{x}) \cdot \sum_{x \in \mathcal{X}}\left|\frac{Z_t}{Z_{t+1}}p_\top^{(t)}(x) - \frac{Z_t^\star}{Z_{t+1}^\star}\pi_t(x)\right|,$$

*in which $\hat{x} \in \arg\max_{x \in \mathcal{X}} f(\mathcal{D}_{t+1}|x)$ is the maximum likelihood instance in $\mathcal{X}$ for $\mathcal{D}_{t+1}$.*

### 4.2 Analysis for KL streaming criterion-based training

We now turn to the case in which SB-GFlowNets are trained with KL streaming updates, sequentially minimizing Equation 5 — in which case we only learn the forward a backward policy networks. More specifically, Proposition 4 provides an upper bound on the TV distance between $p_\top^{(t+1)}$ and $\pi_{t+1}$ as a function of $f(\mathcal{D}_{t+1}|x)$, $p_\top^{(t)}$'s closeness to $\pi_t$, and the learning objective.

**Proposition 4.** *Recall that $p(\tau) \propto p_F^{(t)}(\tau)f(\mathcal{D}_{t+1}|x)$. Thus, under the notations of Proposition 2,*

$$TV\left(p_\top^{(t+1)}, \pi_{t+1}\right) \leq \underbrace{\frac{1}{2}\sqrt{\mathcal{D}_{KL}\left[p_F^{(t+1)}||p\right]}}_{\textit{Estimation error}} + \underbrace{TV\left(\frac{p_\top^{(t)}(\cdot)f(\mathcal{D}_{t+1}|\cdot)}{\mathbb{E}_{y \sim p_\top^{(t)}}[f(\mathcal{D}_t|y)]}, \frac{\pi_t(\cdot)f(\mathcal{D}_{t+1}|\cdot)}{\mathbb{E}_{y \sim \pi_t}[f(\mathcal{D}_t|y)]}\right)}_{\textit{Accuracy of } p_\top^{(t)}}. \quad (7)$$

In spite of considering a different learning objective compared to Proposition 3, Proposition 4 reinforces the importance of properly optimizing the loss in each time step — due to its compounding impact on SB-GFlowNet's accuracy. Nonetheless, the quality of $\pi_t$ is roughly negligible when the observed data set $f(\mathcal{D}_{t+1}|\cdot)$ dominates the shape of $p_\top^{(t)}(\cdot)f(\mathcal{D}_{t+1}|\cdot)$ — and of $\pi_t(\cdot)f(\mathcal{D}_{t+1}|\cdot)$.

## 5 Experiments

We show that SB-GFlowNets can accurately learn the posterior distribution conditioned on the streaming data for one toy and two practically relevant applications. To start with, Section 5.1 illustrates the applicability of SB-GFlowNets in the context of set generation. Nextly, Section 5.2 showcases the correctness of SB-GFlowNets in the problem of Bayesian linear preference learning with integer-valued features [8]. Then, Section 5.3 indicates the potential of SB-GFlowNets for carrying out online Bayesian inference over phylogenetic trees [13, 15, 59]. Finally, Section 5.4 touches on the problem of Bayesian structure learning with streaming data. To the best of our knowledge, SB-GFlowNets are the first method enabling variational inference over discrete parameter spaces within a streaming setting, as previous approaches either relied on intractable continuous relaxations [20, 25, 36] or scaled poorly in the size of the target's support [56]. We provide further implementation details regarding model configurations and experimental settings in Appendix C.

### 5.1 Set generation

**Problem description.** We first remark that, if $R_1, \ldots, R_{k+1}$ are positive functions on $\mathcal{X}$, the problem of streaming Bayesian inference may be generalized to the problem of sampling from

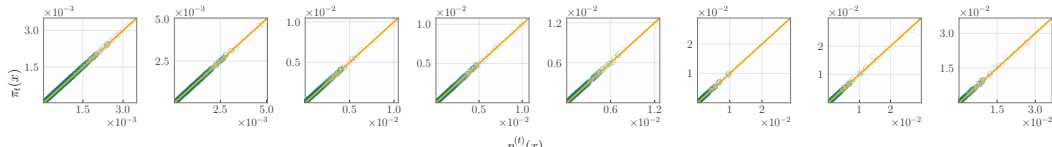

Figure 1: **SB-GFlowNets learn an accurate approximation to a changing target** distribution in a streaming setting for the task of set generation. Each plot depicts the target and learned distributions from the first (left-most) to the last (right-most) streaming update.

$\mathcal{X}$ proportionally to the product $\prod_{i=1}^{k+1} R_i$ by having only access to $R_{k+1}$ and to a GFlowNet approximating $\prod_{i=1}^{k} R_i$. Indeed, in the context Bayesian inference, this correspondence is achieved by defining $R_i(x) := f(\mathcal{D}_i|x)$ for $i > 1$ and $R_1(x) = f(\mathcal{D}_1|x)\pi(x)$. Thus, to demonstrate the effectiveness of SB-GFlowNets and highlight the benefits and pitfalls of each proposed training scheme, we consider the set generation task — a popular toy experiment in the GFlowNet literature [3, 44, 53]. In this task, $\mathcal{X}$ is the space of sets with $S$ elements extracted from a fixed deposit $\mathcal{I} = \{1, \ldots, d\}$. Then, for $i \in \{1, \ldots, K\}$, we define $f_i \colon \mathcal{I} \to \mathbb{R}$ and let $\log R_i(x) = \sum_{e \in x} f_i(e)$ be the $i$th log-reward of a set $x$.

**Experimental setup.** We fix $d = 24$ and $S = 18$. To define $R_i$, we independently sample $f_i(d)$ from an uniform distribution on $[-5, 5]$. Correspondingly, we define $R_i^{(\alpha)} := R_i^{1/\alpha}$ as the tempered reward, noting that $R_i^{(\alpha)}$ becomes increasingly sparse as $\alpha \to 0$.

**Results.** Table 1 highlights the differences between our two training schemes in terms of the target's temperature. For relatively sparse targets (with $\alpha < 1$), the benefits of off-policy sampling enacted by the minimization of the TB loss lead to a better performance relatively to

Table 1: TV between the GFlowNet and the target. TB outperforms KL for sparse targets (small $\alpha$).

| $\alpha$ | 1.00 | 0.75 | 0.50 |
|---|---|---|---|
| TB | $0.21_{\pm 0.06}$ | $0.28_{\pm 0.10}$ | $\mathbf{0.36}_{\pm 0.24}$ |
| KL | $\mathbf{0.13}_{\pm 0.03}$ | $\mathbf{0.17}_{\pm 0.04}$ | $0.55_{\pm 0.38}$ |

the KL-minimizing algorithm. Indeed, the on-policy exploration employed by the latter potentially hinders the model's capabilities of finding the sparsely distributed high-probability regions of the target, slowing down the training convergence and potentially leading to mode collapse [38]. Also, albeit one could implement an IS estimator for off-policy training under the KL criterion, our early experiments suggested that the increased gradient variance outweighed the gains from enhanced exploration. Ultimately, the choice of an appropriate learning objective for training GFlowNets will depend on the application. For training SB-GFlowNets, Figure 1 shows that both TB and KL minimization result in accurate approximations to the changing posterior.

### 5.2 Linear preference learning with integer-valued features

**Problem description.** Consider a collection of instances $\{y_i\}$ endowed with a transitive and complete preference relation $\succeq$; we assume that each $y_i \in \{1, 0\}^d$ is a binary feature vector. Naturally, the preference relation $\succeq$ is represented as a mapping $u$ such that $y \succ y'$ if and only if $u(y) > u(y')$; uncovering $u$ is the major goal of preference learning methods. Similarly to [8, 21], we here assume that $u$ is a linear function, $u(y) = x^T y$, for a integer-valued vector $x$ and that we have access to a data set $\mathbf{y} = \{(y_{i1}, y_{i2}, p_i)\}_i$ denoting whether $y_{i1}$ is preferred to $y_{i2}$ ($p_i = 1$) or otherwise ($p_i = 0$) for a fixed individual. Subsequently, we define the probabilistic model

$$p(p_i = 1|x, (y_{i1}, y_{i2})) := \sigma(u(y_{i1}) - u(y_{i2})) = \sigma\left(x^T(y_{i1} - y_{i2})\right), \tag{8}$$

in which $\sigma$ is the sigmoid function, and a prior distribution $\pi(x)$ over $x$ [18]; the intuition is that a larger difference between $y_{i1}$ and $y_{i2}$'s utilities makes the event in which $y_{i1}$ is preferred over $y_{i2}$ more likely. Our goal is to infer the individual's preferences based on the posterior $\pi(x|\mathbf{y})$ for some unseen pair $(\tilde{y}_1, \tilde{y}_j)$, i.e., to estimate the predictive distribution $p(\tilde{y}|(\tilde{y}_1, \tilde{y}_2), \mathbf{y})$.

**Experimental setup.** We assume $x \in [[0, 4]]^d$ and $d = 24$. At each iteration of the streaming process, we sample a novel subset of the $2^{d-1} \cdot (2^d - 1)$ pairs of $d$-dimensional binary feature vectors and use them to update the GFlowNet. The prior on $x$ is a factorized truncated Poisson with $\lambda = 3$.

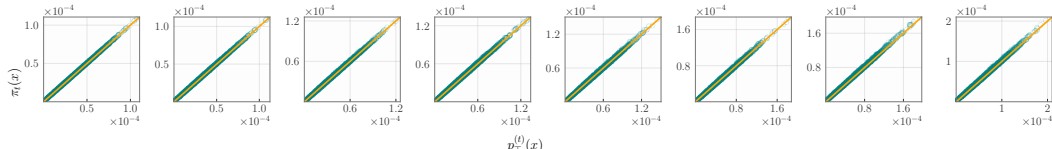

Figure 2: **SB-GFlowNet accurately learns the posterior over the utility's parameters in a streaming setting.** Each plot compares the marginal distribution learned by SB-GFlowNet (horizontal axis) and the targeted posterior distribution (vertical axis) at increasingly advanced stages of the streaming process, i.e., from $\pi_1(\cdot|\mathcal{D}_1)$ (left-most) to $\pi_8(\cdot|\mathcal{D}_{1:8})$ (right-most).

**Results.** Figure 2 shows SB-GFlowNet correctly samples form the posterior in a streaming setting. Note, in particular, that the GFlowNet maintains a high distributional accuracy throughout the streaming iterations. Correspondingly, the predictive log-likelihood of a fixed held out data set monotonically increases as a function of the amount of data consumed by the SB-GFlowNet, as shown in Figure 3; this behavior, also exhibited by the true posterior [55], emphasizes the similarity between the learned and targeted distributions.

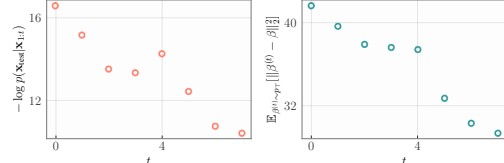

Figure 3: Predictive performance of SB-GFlowNets in terms of pred. NLL and avg. MSE. SB-GFlowNets behaves similarly to the ground-truth, wrt how the NLL evolves as a function of data chunks.

### 5.3 Online Bayesian phylogenetic inference

**Problem description.** Bayesian phylogenetic inference (BPI) [29, 60] aims to infer structural properties of evolutionary trees given molecular sequences such as DNA and RNA. Formally, let $\{S_1, \ldots, S_N\}$ be a collection of $N$ DNA sequences of size $M$, $S_i \in \{A, T, C, G\}^M$, one for each biological species. We define a *phylogeny* as a tuple $T = (t, \mathbf{b})$ comprising a *rooted tree topology* $t$ and its non-negative *branch lengths* $\mathbf{b}$. In this context, a rooted tree topology is a complete binary tree having $N$ labeled leaves corresponding to the considered species and $N - 1$ unlabeled internal nodes corresponding to their ancestors. To carry out Bayesian inference over the space of phylogenies, we adopt Jukes & Cantor's nucleotide substitution model (JC69; [26]) to define a observational model over the observed DNA sequences; to calculate the likelihood, we use Felsenstein's algorithm [15].

Crucially, despite the popularity of BPI methods, the development of new sequencing technologies has swiftly led to the enlargement of already sizeable sequence databases. In this scenario, maintaining an up-to-date estimate of the posterior became an increasingly difficult task due to the necessity of re-estimating the full posterior from scratch every time a new batch of data is collected. In the following, we show that GFlowNets, which have recently shown SOTA performance in BPI [64], can also accurately update the posterior on phylogenetic trees given additional sequences.

**Experimental setup.** We generate the data by simulating JC69's model for a collection of $N = 7$ species and a substitution rate of $\lambda = 5 \cdot 10^{-1}$ (see [59], Chapter 1). At each iteration, we sample a new DNA subsequence of size $10^2$ for each species and update SB-GFlowNet according to Algorithm 1. For Table 2, $|\mathcal{D}_1| = 10^3$ and $|\mathcal{D}_2| = 10^2$. The prior is an uniform distibution over phylogenies.

**Results.** Figure 4 (left) shows that the learned posterior distribution gets increasingly concentrated on the true phylogenetic tree; this behavior, which is inherent to posteriors over phylogenies under uniform priors [50], emphasizes the similarity between the learned and targeted distributions for SB-GFlowNets. Figure 4 (right), on the other hand, also suggests that the model's accuracy decreases as a function of the number of streaming updates. Nonetheless, we believe that this is predominantly caused by the poste-

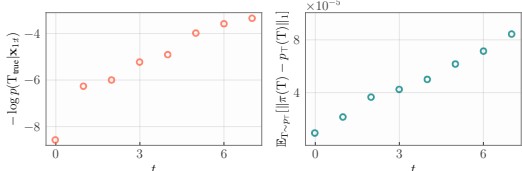

Figure 4: SB-GFlowNet's accurate fit to the true posterior in terms of the probability of the true phylogeny (left) and of the learned model's accuracy (right).

rior's increasing sparsity due to the expanding data sets [50], which makes it more difficult for the GFlowNet to learn a good approximation [11], instead of by an intrinsic limitation of SB-GFlowNets. An investigation of this phenomenon and of the conditions upon which re-train the SB-GFlowNet

Table 2: **SB-GFlowNet significantly accelerates the training of GFlowNets** in a streaming setting. Indeed, SB-GFlowNets achieve an accuracy comparable to a GFlowNet trained from scratch to sample from $\pi_2(\cdot|\mathcal{D}_{1:2})$ in less than half the time (measured in seconds per 20k epochs).

| Model | Number of leaves | | |
|---|---|---|---|
| | 7 | 9 | 11 |
| GFlowNet | 2846.88 $s$ | 3779.11 $s$ | 4821.74 $s$ |
| SB-GFlowNet (*ours*) | **1279**.68 $s$ | **1714**.49 $s$ | **2303**.99 $s$ |
| Relative accuracy gain (TV) | 0.00±0.04 | −0.02±0.04 | 0.00±0.01 |

based on an earlier checkpoint due to the accumulated unreliability of the streaming updates, as described by Proposition 4, is left application-dependent and is thereby left to future endeavors. Finally, Table 2 shows that SB-GFlowNets, which avoids evaluating the full posterior, more than halve the training time required by a standard GFlowNet, while achieving a similar performance in terms of the total variation ($TV$) distance between the learned and target distributions.

## 5.4 Bayesian structure learning

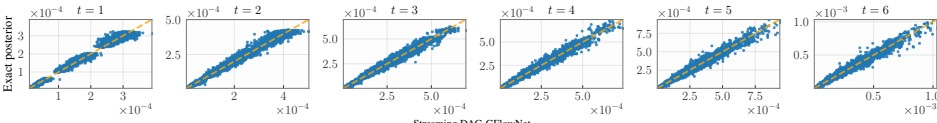

Figure 5: **SB-GFlowNets accurately learns a distribution over DAGs** for causal discovery in each time step. At each update, an additional dataset of 200 points was sampled from the true model. For this problem, we implemented a DAG-GFlowNet on 5-variable data sets, similarly to [11, Figure 3].

**Problem description.** Let $\mathbf{X} \in \mathbb{R}^{n \times d}$ be a data set distributed according to a linear Gaussian structural equation model (SEM), i.e., $\mathbf{X} = \beta\mathbf{X} + \epsilon$, in which $\beta \in \mathbb{R}^{d \times d}$ is a (sparse) matrix and $\epsilon \in \mathbb{R}^{n \times d}$ are $nd$ i.i.d. samples from a normal distribution. We assume that the adjacency matrix induced by $\beta$, $[\beta \neq 0] \in \{1, 0\}^{d \times d}$, characterizes a directed acyclic graph on the dataset's variables. In this case, the linear Gaussian SEM represents a Bayesian network, and it is the goal of *Bayesian structure learning* algorithms to find such a network based on the observed $\mathbf{X}$ [11]. Although this problem has been studied beyond the constraints of linear models governed by Gaussian distributions [12], we focus on a simplified setting in this section. In particular, given a sequence $\{\mathbf{X}_t\}_{t \geq 1}$ of i.i.d. realizations of a linear Gaussian SEM, we define a belief distribution over Bayesian networks $\mathcal{N} = (\{1, \ldots, d\}, \mathbf{A})$ on variables $\{1, \ldots, d\}$ and adjacency matrix $\mathbf{A}$ as

$$\log R_t(\mathcal{N}) = \max_{\beta : [\beta \neq 0] = \mathbf{A}} \ell(\beta|\mathbf{X}_t) \text{ and } R_{1:t}(\mathcal{N}) = \prod_{1 \leq i \leq t} R_t(\mathcal{N}), \tag{9}$$

in which $\ell(\beta|\mathbf{X}_t)$ is the log-likelihood of $\mathbf{X}_t$ under the linear Gaussian SEM parameterized by $\beta$. SB-GFlowNets can naturally handle streaming inference for this model by casting each $R_t$ as a subposterior distribution — in the same fashion of the set generation task in Section 5.1.

**Experimental setup.** Likewise Deleu et al. [11, Figure 5], we evaluate SB-GFlowNets on models with $d = 5$ variables. Also, we sample a new 200-sized data set from a fixed linear Gaussian SEM for each streaming update of the GFlowNet. To ensure that the generated samples are valid DAGs, we adopt Deleu et al. [11]'s DAG-GFlowNet.

**Results.** Our results corroborate the findings of Section 5.3. On the one hand, Figure 5 highlights that our SB-GFlowNet can accurately learn a distribution over DAGs across a range of streaming updates. On the other hand, Figure 5 shows that the probability mass assigned by the SB-GFlowNet to the true DAG responsible for the data-generating process increases as more samples are incorporated into the belief distribution. Hence, SB-GFlowNets abide by the desiradata of a streaming algorithm for Bayesian inference.

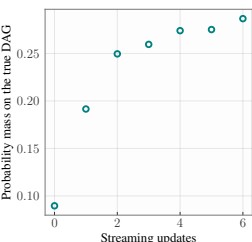

Figure 6: The probability mass on the true DAG increases as more samples are added to SB-GFlowNet.

# 6 Related works

**Applications of GFlowNets.** GFlowNets [2, 3, 30] were originally proposed as reinforcement learning algorithms to train a stochastic policy to sample states proportionally to a prescribed reward function, From this point on, GFlowNets were widely applied to problems as diverse as probabilistic modeling [23, 61], combinatorial optimization [62], Bayesian causal discovery [11, 12] and phylogenetic inference [64], symbolic regression [32], stochastic control [30], language modeling [22], and Bayesian deep learning [34]. Recently, there is an increasing literature concerned with the possibility of composing multiple pre-trained GFlowNets to fit them to specific applications [17] and to accelerate training convergence [31]. In this context, we show how the composition of two GFlowNets through the SB condition leads to an efficient learning algorithm for Bayesian inference under a streaming setting.

**Streaming Variational Bayes.** Streaming, Distributed, Asynchronous (SDA) Bayes [6] was originally presented as a general framework for implementing variational inference algorithms in a streaming setting. Similarly to our work, it was based upon the principle that, if the variational distribution $q^{(t)}(x)$ is a good approximation to the unnormalized posterior $\tilde{\pi}_t(x)$, then $q^{(t+1)}(x)$ may be optimized to approximate $q^{(t)}(x)f(\mathcal{D}_t|x)$ instead of $f(\mathcal{D}_t|x)\tilde{\pi}_t(x)$. This framework was instantiated to accommodate, for example, the learning of Gaussian processes [7], of tensor factorization [14], of feature models [51] and, jointly with SMC, nonlinear state-space models [63]. Nonetheless, our work is the first one to enable the training of GFlowNets in a streaming setting and, due to the relationship between GFlowNets and variational inference algorithms presented by [38, 65], may be viewed as an instance of SDA-Bayes tailored to inferential problems on a combinatorial support. In the realm of approximate Bayesian inference, Akhound-Sadegh et al. [1], Berner et al. [4], Sendera et al. [52], Zhang and Matsen [60] propose advancements in diffusion-based sampling methods, which we believe could be extended to the context of approximate streaming Bayesian inference via the techniques presented in this work. On the other hand, Mittal et al. [40] introduces an neural network-based approach for approximate Bayesian inference that amortizes over exchangeable data sets to handle posterior inference in novel data. Also, Richter and Berner [48], Richter et al. [49] develop a low-variance gradient estimator for carrying out variational inference derived from the *log-variance loss*, which could be employed as an alternative to our KL-based objective for streaming updates of GFlowNets. On a broader scale, Cranmer et al. [9] reviews approximate Bayesian methodology under the lens of simulation-based inference.

# 7 Conclusions, limitations, and outlook

**Conclusions.** We proposed the first method for carrying out approximate Bayesian inference over discrete distributions within a streaming setting, called SB-GFlowNet. We proposed two training/update schemes for SB-GFlowNet, as well as a theoretical analysis accounting for the compounding effect of errors due to posterior propagation. Our experimental evaluation showcases SB-GFlowNet's effectiveness in accurately sampling from the target posterior — while still achieving a significant reduction in training time relative to standard GFlowNets.

**Limitations.** Proposition 2 and Proposition 4 suggest that an inappropriate approximation to $\pi_t$ may be propagated through time and lead to increasingly inaccurate models. This phenomenon, known as *catastrophic forgetting* in the online literature [19, 39], may eventually demand retraining of the current SB-GFlowNet based on an earlier checkpoint or on the full posterior. We also note that unlike traditional variational methods, where the expressiveness of the approximation family is explicitly chosen, the expressiveness implied by different parametrizations of GFlowNets is not clear. We believe this is a fruitful avenue for future investigation.

**Outlook and future works** As our proposal provides generic and efficient streaming variational inference solution for discrete parameters, we believe it will allow the use of streaming discrete Bayesian inference to more datasets and to less explored research domains. In this work, we proposed two training criteria based on the trajectory balance condition and the KL criterion, nonetheless, these are not the only training schemes for GFlowNets, we believe other proposed criteria such as *detailed balance* [3] or the *sub-trajectory balance* [35] could be adapted for the streaming setting as well. Additionally, due to the flexibility of GFlowNets in sampling unnormalized distributions, our proposal can be extended to different divergences and generalized likelihoods to improve the predictive quality of the learned posteriors [28]. Finally, the problem of updating a GFlowNet when the size of the generated object changes, e.g., when a new species is observed during BPI, remains open.

## Acknowledgements

This work was supported by the Fundação Carlos Chagas Filho de Amparo à Pesquisa do Estado do Rio de Janeiro FAPERJ (SEI-260003/000709/2023), the São Paulo Research Foundation FAPESP (2023/00815-6), the Conselho Nacional de Desenvolvimento Científico e Tecnológico CNPq (404336/2023-0), and the Silicon Valley Community Foundation through the University Blockchain Research Initiative (Grant #2022-199610).

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

---

**Algorithm 1** Training a SB-GFlowNet by minimizing $\mathcal{L}_{SB}$

---

**Require:** $(\mathcal{D}_t)_{T \geq t \geq 1}$ streaming data sets, $f(\cdot|x)$ a likelihood model parametrized by $x$, $\pi(x)$ a prior distribution over $\mathcal{X}$

**Ensure:** $G_T$ samples proportionally to $\left(\prod_{t=1}^{T} f(\mathcal{D}_t|x)\right)\pi(x)$

$\quad G_1 \leftarrow (p_F^1, p_B^1, Z_1) := \underset{p_F, p_B, Z}{\arg\min} \, \mathbb{E}_{\tau \sim \xi}\left[\mathcal{L}_{TB}(\tau; p_F, p_B, Z)\right] \qquad \triangleright$ Roughly minimize $\mathcal{L}_{TB}$ via SGD

$\quad$ **for** $t$ in $\{2, \ldots, T\}$ **do** $\qquad\qquad\qquad\qquad\qquad\qquad\qquad\qquad\quad \triangleright$ Roughly minimize $\mathcal{L}_{SB}$ via SGD

$\qquad G_t \leftarrow (p_F^{(t)}, p_B^{(t)}, Z_t) := \underset{p_F, p_B, Z}{\arg\min} \, \mathbb{E}_{\tau \sim p_\top^{(t-1)}}\left[\mathcal{L}_{SB}(\tau; p_F, p_B, Z; G_{t-1})\right]$

$\quad$ **end for**

---

## A Training of SB-GFlowNets

Algorithm 1 outlines the training of a SB-GFlowNet by minimizing the SB loss. As we described in the text, the first GFlowNet is trained conventionally by approximately minimizing either the TB loss or the KL divergence between the forward and backward policies. Then, the subsequent models are trained by the approximate minimization of either the SB loss or streaming balance criterion. Notably, the problem of streaming update may be framed as the learning of GFlowNets with stochastic rewards [45], which are not determistically associated to the terminal states.

## B Proofs

### B.1 Proof of Proposition 1

We are assuming that $p_\top^{(t)} \propto \pi_t$ and that $\mathbb{E}_{\tau \sim \xi}[\mathcal{L}_{SB}(\tau)] = 0$ for a distribution $\xi$ of full support. Thus, $\mathcal{L}_{SB}(\tau) = 0$ for all $\tau$. As a consequence,

$$p_F^{(t+1)}(\tau) = \left(\frac{p_B^{(t+1)}(\tau|x)}{p_B^{(t)}(\tau|x)}\right) \cdot \frac{Z_t}{Z_{t+1}} \cdot p_F^{(t)}(\tau) f(\mathcal{D}_{t+1}|x). \tag{10}$$

By assumption, $(p_F^{(t)}, p_B^{(t)}, Z_t)$ satisfy the trajectory balance condition with respect to $\pi_t$, therefore, $Z_t \cdot p_F^{(t)}(\tau) = p_B^{(t)}(\tau|x)\pi_t(x)$. Thus,

$$\frac{p_F^{(t+1)}(\tau)}{p_B^{(t+1)}(\tau|x)} = Z_{t+1}\pi_t(x)f(\mathcal{D}_{t+1}|x). \tag{11}$$

Finally, by summing over $\tau \rightsquigarrow x$:

$$p_\top^{(t+1)}(x) := \mathbb{E}_{\tau \sim p_B^{(t+1)}(\cdot|x)}\left[\frac{p_F^{(t+1)}(\tau)}{p_B^{(t+1)}(\tau|x)}\right] \propto \pi_t(x)f(\mathcal{D}_{t+1}|x) \tag{12}$$

$$\propto \pi_{t+1}(x). \tag{13}$$

### B.2 Proof of Proposition 2

Firstly, we note that $\delta_{LS}^\xi(p, q)$ is a metric. Thus, by the triangle inequality,

$$\delta_{LS}^{\pi_{t+1}}\left(p_\top^{(t+1)}, \pi_{t+1}\right) \leq \delta_{LS}^{\pi_{t+1}}\left(p_\top^{(t+1)}, \hat{p}_\top^{(t+1)}\right) + \delta_{LS}^{\pi_{t+1}}\left(\hat{p}_\top^{(t+1)}, \pi_{t+1}\right). \tag{14}$$

The first term of the right-hand-side of the preceding equation corresponds to the estimation error associated to the GFlowNet's learning problem. For the second term, note that $\hat{p}_\top^{(t+1)}(x) = \frac{Z_t}{\hat{Z}_{t+1}}p_\top^{(t)}(x)f(\mathcal{D}_{t+1}|x)$ by assumption ($\hat{p}_\top^{(t+1)}$ satisfies the SB condition). Thus,

$$\log \hat{p}_\top^{(t+1)}(x) - \log \pi_{t+1}(x) = \log \frac{Z_t}{\hat{Z}_{t+1}} + \log p_\top^{(t)}(x)f(\mathcal{D}_{t+1}|x) - \log \frac{Z_t^\star \pi_t(x)}{Z_{t+1}^\star}f(\mathcal{D}_{t+1}|x)$$

$$= \log \frac{Z_{t+1}^\star}{\hat{Z}_{t+1}} + \log \frac{Z_t}{Z_t^\star} + \log \frac{p_\top^{(t)}(x)}{\pi_t(x)}.$$

The result follows by a further application of the triangle inequality to $\delta_{LS}^{\pi_{t+1}}\left(p_{\top}^{(t+1),\star}, \pi_{t+1}\right)$, namely,

$$
\delta_{LS}^{\pi_{t+1}}\left(\hat{p}_{\top}^{(t+1)}, \pi_{t+1}\right) = \mathbb{E}_{x \sim \pi_{t+1}}\left[\left(\log \frac{Z_{t+1}^{\star}}{\hat{Z}_{t+1}} + \log \frac{Z_t}{Z_t^{\star}} + \log \frac{p_{\top}^{(t)}(x)}{\pi_t(x)}\right)^2\right]^{1/2}
$$
$$
\leq \left|\log \frac{\hat{Z}_{t+1}}{Z_{t+1}^{\star}}\right| + \left|\log \frac{Z_t}{Z_t^{\star}}\right| + \delta_{LS}^{\pi_{t+1}}\left(p_{\top}^{(t)}, \pi_t\right). \tag{15}
$$

Proposition 2 is obtained by plugging Equation 15 into Equation 14.

### B.3   Proof of Proposition 3

This result, which follows from reasoning similar to the one in Proposition 2 above, aims to show the dependence of the model's performance on the newly observed dataset. In this sense, notice that

$$
TV\left(p_{\top}^{(t+1)}, \pi_{t+1}\right) \leq TV\left(p_{\top}^{(t+1)}, \hat{p}_{\top}^{(t+1)}\right) + TV\left(\hat{p}_{\top}^{(t+1)}, \pi_{t+1}\right). \tag{16}
$$

Thus, since $\hat{p}_{\top}^{(t+1)}(x) = \frac{Z_t}{\hat{Z}_{t+1}} p_{\top}^{(t)}(x) f(\mathcal{D}_{t+1}|x)$,

$$
TV\left(\hat{p}_{\top}^{(t+1)}, \pi_t\right) = \frac{1}{2}\sum_{x \in \mathcal{X}}\left|p_{\top}^{(t+1)}(x) - \pi_{t+1}(x)\right|
$$
$$
= \frac{1}{2}\sum_{x \in \mathcal{X}} f(\mathcal{D}_{t+1}|x)\left|\frac{Z_t}{\hat{Z}_{t+1}} p_{\top}^{(t)}(x) - \frac{Z_t^{\star}}{Z_{t+1}^{\star}} \pi_t(x)\right| \tag{17}
$$
$$
\leq \frac{1}{2} f(\mathcal{D}_{t+1}|\hat{x})\sum_{x \in \mathcal{X}}\left|\frac{Z_t}{\hat{Z}_{t+1}} p_{\top}^{(t)}(x) - \frac{Z_t^{\star}}{Z_{t+1}^{\star}} \pi_t(x)\right|.
$$

The result follows by plugging Equation 17 into Equation 16.

### B.4   Proof of Proposition 4

This result follows from the successive application of the triangle, Pinsker's [10] and Jensen's inequality applied to the KL divergence. More specifically, first note that the optimal distribution under the KL streaming criterion satisfies $\hat{p}_{\top}^{(t+1)} \propto p_{\top}^{(t)} f(\mathcal{D}_{t+1}|x)$. Then, by the triangle inequality,

$$
TV\left(p_{\top}^{(t+1)}, \pi_{t+1}\right) \leq TV\left(p_{\top}^{(t+1)}, \hat{p}_{\top}^{(t+1)}\right) + TV\left(\hat{p}_{\top}^{(t+1)}, \pi_{t+1}\right). \tag{18}
$$

For the first term, note that

$$
TV\left(p_{\top}^{(t+1)}, \hat{p}_{\top}^{(t+1)}\right) \leq \frac{1}{2}\sqrt{\mathcal{D}_{KL}\left[p_{\top}^{(t+1)}||\hat{p}_{\top}^{(t+1)}\right]} \leq \frac{1}{2}\sqrt{\mathcal{D}_{KL}\left[p_F^{(t+1)}||p\right]}, \tag{19}
$$

since $\hat{p}_F^{(t+1)} \propto p$ by definition; recall that $p(\tau) = p_F^{(t)}(\tau) f(\mathcal{D}_{t+1}|x)$. Here, the first inequality follows from Pinsker's inequality and the second one, from the data-processing inequality. For the second term, note that

$$
\hat{p}_{\top}^{(t+1)}(x) = \frac{p_{\top}^{(t)}(x) f(\mathcal{D}_{t+1}|x)}{\sum_{y \in \mathcal{X}} p_{\top}^{(t)}(y) f(\mathcal{D}_{t+1}|y)} = \frac{p_{\top}^{(t)}(x) f(\mathcal{D}_{t+1}|x)}{\mathbb{E}_{y \in p_{\top}^{(t)}}\left[f(\mathcal{D}_{t+1}|x)\right]}, \tag{20}
$$

with a corresponding representation of $\pi_{t+1}$ as a function of $\pi_t$ and $f(\mathcal{D}_{t+1}|x)$. The result follows by pluggin Equation 20 and Equation 19 into Equation 18.

## C   Experimental details

We provide below details for reproducing our experiments for each considered generative task. To approximately solve the optimization problem outlined in Algorithm 1, we employed the Adam optimizer [27] with a learning rate of $10^{-3}$ for the $p_F$'s parameters and $10^{-1}$ for $\log Z_t$, following recommendations from [37]. Also, we linearly decreased the learning rate during training. Experiments were run in a cluster equipped with A100 and V100 GPUs, using a single GPU per run.

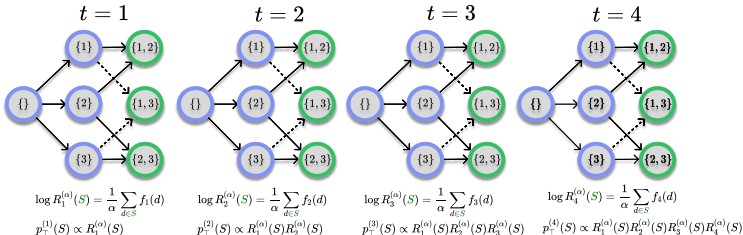

Figure 7: **Illustration of the task of generating sets** of size $|S| = 2$ with elements in $\{1, 2, 3\}$. On each streaming update, a novel reward function $R_i^{(\alpha)}$ is observed; a small value of $\alpha$ entails a more sparse and harder-to-sample-from distribution. Terminal states $\mathcal{X}$ are illustrated in green and non-terminal states are depicted in blue. At the $t$th iteration, we learn a generative model $p_\top^{(t)}$ sampling $S \in \mathcal{X}$ proportionally to $\prod_{1 \le i \le t} R_t^{(\alpha)}(S)$.

### C.1 Set generation

**Experimental setup.** We fixed $d = 24$ and $S = 18$. To parameterize the forward policy, we implemented an two-layer neural network with a 128-dimensional latent embedding. For the streaming updates, we fixed $\alpha = 1$ and randomly sampled the objects' utilities at each novel iteration. We trained the model by minimizing the KL streaming criterion.

**GFlowNet's design.** To generate a set $x \in \mathcal{X}$, we iteratively add elements randomly sampled from $\mathcal{I}$ to an initially empty $x$ until $x$ has size $S$; see Figure 7. The forward policy is parameterized as a two-layer neural network and the backward policy is fixed as an uniform distribution.

### C.2 Linear preference learning with integer-valued features

**Experimental setup.** We assume that $x \in [[0, 4]]^d$ and $d = 24$ and that the data was simulated from the observational model. At each streaming round, a novel and independent data set was simulated and the model was trained by minimizing the SB loss. To parameterize the forward policy, we implemented an MLP with 2 64-dimensional layers receiving the padded parameter $x$ as an input and returning a probability distribution over $[[0, 4]]$.

**GFlowNet's design.** The generative process implemented by the GFlowNet consists of, starting at an initially empty state $x_o$, iteratively sampling a value from $[[0, 10]]$ and appending this value to the current state until its length reaches $d$. To parameterize the forward policy, we use an MLP with two 64-dimensional layers that receives the padded state as a fixed-size input.

### C.3 Online Bayesian phylogenetic inference

**Experimental setup.** We assume the observational data follow the J&C69 mutation model with an instantaneous mutation rate of $\lambda = 5 \cdot 10^{-3}$. The data was synthetically generated from the corresponding observational model conditioned on a randomly sampled tree with 7 leaves, and this process was repeated at each streaming round. To illustrate the computational gains enacted by the implementation of SB-GFlowNets in Table 2, we considered updating a GFlowNet trained on an initially large data set according to the newly observed and relatively small biological sequences, which is a common challenge faced by practitioners. Then, by avoiding the additional log-likelihood evaluations, we achieved significantly faster inference.

**GFlowNet's design.** A state consists of a forest of complete binary trees. Initially, all leaves are roots of their own singleton trees and, at each iteration of the generative process, we select two trees and join their roots to a newly added unlabelled node. This procedure is finished when all leaves are connected; see [64, Figure 1] for an illustration of this generative process. To parameterize $p_F$, we use a graph isomorphism network (GIN; [57]); the backward policy is fixed as uniform.

### C.4 Bayesian structure learning

**Experimental setup.** We sample each component of the error terms $\epsilon$ from a zero-centered Gaussian distribution with standard deviation $\sigma = 5 \cdot 10^{-2}$. Also, we select a random Bayesian network from a directed configuration model [43] and draw the components $\beta$ from a corresponding standard Gaussian distribution to define the true data-generating process. To parameterize the policy network, we use an MLP with 2 128-dimensional layers receiving the DAG's flattened adjacency matrix as input.

**GFlowNet's design.** We adopt Deleu et al. [11]'s DAG-GFlowNet. In a nutshell, the generative process starts at an edgeless graph and each transition either adds an edge to the current state or triggers a stop. To ensure the acyclicity of the generated samples, we follow [11, Appendix C] and iteratively update a binary vector $\mathbf{m} \in \{1,0\}^{d \times d}$ indicating which edges in the adjacency matrix can be safely added to the current state without forming cycles.

## D    On the permutation invariance of SB-GFlowNets

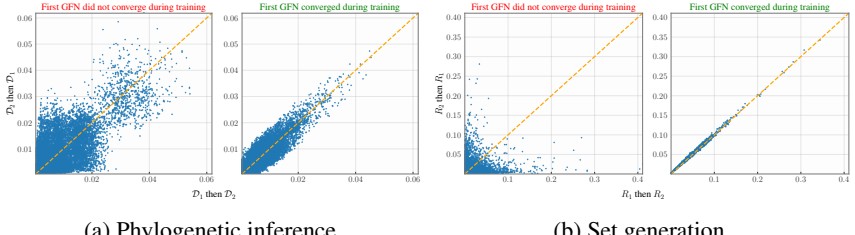

(a) Phylogenetic inference.                    (b) Set generation.

Figure 8: **Permutation invariance of SB-GFlowNets** for phylogenetics (a) and set generation (b). When the first GFlowNet is not adequately trained, the learned distribution after two streaming updates depends on the ordering of the observed datasets (left (a), left (b)). In contrast, when both the first and second GFlowNets are accurate, the resulting distribution is approximately invariant to the data permutation (right (a), right (b)).

Exchangeability is a natural property of each case study we presented in the main text. We thus ask: are SB-GFlowNets permutation invariant? Clearly, the distribution learned by a SB-GFlowNet does not depend on the order in which the data sets are observed when the SB condition is satisfied at each streaming update; this is a direct consequence of Proposition 1. On the other hand, permutation invariance is not guaranteed when the learning objectives are only imperfectly minimized; see Figure 8 for an extreme example. To the best of our knowledge, however, this sensibility to data ordering is a property of every approximate streaming Bayesian inference method, e.g., [6, 13].

