# OpenReview forum: "Streaming Bayes GFlowNets"
_NeurIPS.cc/2024/Conference — NeurIPS 2024 poster_

### Official Review · Reviewer_bMoM · 2024-07-12

**Soundness:** 4
**Presentation:** 3
**Contribution:** 3
**Rating:** 7
**Confidence:** 4

**Summary:**

Generative flow nets are extended to streaming inference by using the prediction of the previous network as the prior for the current one. Two training methods are proposed: streaming balance which uses squared difference of target and learned log posteriors, and VI which uses KL divergence. Analytic results for both methods bound the error in terms of updating error and error from the previous step. Three experiments show the method can accurately track the true posterior across updating steps.

**Strengths:**

The methods are elegant and sound and the experiment results are strong. This is an important alternative to streaming VI that I'm excited to try out myself.

**Weaknesses:**

Not a major weakness, but the extension to the streaming case (this paper's main contribution) is fairly straightforward. It was clear from the beginning that the predicted distribution from the previous step would provide the prior for the next, with the obvious adaptations of the balance and VI objectives.

Both $p_B$ terms in the streaming balance condition (3) appear to be unnecessary. Prop 1 will hold without them. We just need a joint distribution over $(x,\tau)$ (equivalently, a distribution over $\tau$) with the desired marginal on $x$, and $f(\mathcal{D}_{t+1}|x) p^t_F(\tau)$ already has that property. Compare to the definition of the target distribution $p(\tau)$ in (5).

Sec 3.2 just gives the objective and doesn’t explain how $p_F$ and $Z$ are learned (likewise with the argmax lines in Algo 1). Is it just SGD on $\mathcal{L}_{SB}$?

The main text says nothing about how to construct the graph ($\mathcal{S}$ and $\boldsymbol{A}$). Details are given in the appendix -- in all cases the allowable paths build the desired objects from simpler ones -- but a sentence in the main text would help.

**Questions:**

(2): why do we need to sample from $p_B$ and importance weight, rather than sampling directly from $p_F$?

(5),170,178: $\pi_{t+1}$ should be $f(\mathcal{D}_{t+1}|x)$, yes? The derivations all look correct if so.

181: what is $\delta$?

$Z_{(t+1)}$ at 205 and $\hat{Z}_{t+1}$ are the same?

More of a suggestion: could the method be extended to cases where the target objects grow at each time step? For example, phylogenetic trees would grow as new species are observed. It seems like a natural application of flow nets since you can expand the graph by appending a new terminal layer.

**Limitations:**

The set generation task is degenerate because the objective is additive. A model only needs to track the posterior on individual $x$s.

---

> ### Author Rebuttal · Authors · 2024-08-02
>
> Thank you for carefully reading and appreciating our work. Below, we address each of your comments.
>
> > Both $𝑝\_{𝐵}$ terms in the streaming balance condition (3) appear to be unnecessary.
>
> Indeed, when $p_{B}^{t}(\cdot | s)$ is fixed, it does not depend on $t$ and the terms corresponding to the backward policy in Equation (3) cancel out. However, albeit relying on a fixed $p_{B}$ is an often implemented strategy for learning GFlowNets — that we adopt in our work —, $p_{B}$ can in principle be jointly learned with $p_{F}$. In this case, $p_{B}^{t}$ would also depend on $t$ and there would be no cancellations in Equation (3). Nonetheless, due to the lack of clear empirical evidence in the literature supporting the learning of $p_{B}$, we choose to fix it in all our experiments. We will emphasize this point and include the simplified version of Equation (3) in the revised manuscript.
>
> > Sec 3.2 just gives the objective and doesn’t explain how $𝑝\_{𝐹}$ and $𝑍$ are learned
>
> Thanks for noticing. You are correct: both $p_{F}$ and $Z$ in Section 3.2 and in Algorithm 1 are learned by minimizing the corresponding objectives via SGD. We will point this out in the updated manuscript.
>
> > The main text says nothing about how to construct the graph ($𝑆$ and $𝐴$). Details are given in the appendix -- in all cases the allowable paths build the desired objects from simpler ones -- but a sentence in the main text would help.
>
> Thank you for the suggestion. Indeed, we briefly discuss this in the lines 81-84 of section 2 preliminaries and in more detail in the appendix, but we will make this more explicit in the revised manuscript.
> The design of the graph ($S,A$) is application-dependent and should be considered on a problem-by-problem basis. As a general guideline, when the elements of the target’s support $\mathcal{X}$ can be described as being built from simpler partial objects, then, the initial state can correspond to an empty object and the non-terminal states would be partial compositions with an edge from $s$ to $s’$ meaning that $s’$ differs from $s$ by a single additional atomic component. For example, a phylogeny is a rooted tree where all the leaves correspond to the species observed and the internal nodes correspond to unobserved ancestors. The states in the support are single trees and the non-terminal states are forests, i.e. disconnected collections of trees, where the construction follows by joining two trees in the forest by an unobserved node. In other words, the state graph is implicitly defined by iteratively applying actions to the initial state.
>
>
> > (2): why do we need to sample from $p_{B}$
>
> In fact, we could directly sample from $p_{F}$ and count the relative frequency of $x$ to estimate the marginal $p_{T}(x)$. However, our experience suggests that the estimator in Equation (2), which is also used by Malkin et al. [1] and Zhout et al. [2], generally yields better estimates of $p_{T}(x)$.
>
> > (5), 170, 178:  $\pi_{𝑡+1}$ should be $𝑓(\mathcal{𝐷}_{𝑡+1}|𝑥)$, yes?
>
> Yes! Thank you for the observation. We will fix this in the revised manuscript.
>
> > 181: what is $\delta$?
>
> This is a typo; we meant $\nabla\_{\theta} \mathbb{E}\_{\tau \sim p_{F}^{t + 1}}[\gamma(\tau)]$ instead of $\mathbb{E}[\delta(\tau)]$.
>
> > $𝑍_{(𝑡+1)}$ at 205 and $\hat{𝑍}_{𝑡+1}$ are the same?
>
> Yes! This too is a typo.
>
> > More of a suggestion: could the method be extended to cases where the target objects grow at each time step?
>
> Thank you for the question. We do not believe that SB-GFlowNets can be immediately extended to accommodate the training of GFlowNets when the size of the generated objects grows. There are two main reasons for this. Firstly, the neural network that encodes the policy often receives a fixed-size input corresponding to a representation of the current state. When the size of this representation changes, it is unclear how the neural network should be updated. However, for many applications, this could be circumvented by using GNN- or DeepSet-based policy networks, which work well with inputs of varying size. Secondly, it is not totally clear what would happen with the terminal flows when additional nodes/transitions are added to the flow network. Which parts of the network remain balanced upon this addition? Should we learn a novel policy on the expanded network, or can we exploit the knowledge of past models to alleviate the learning problem? These questions are both important and interesting, deserving a work of their own.
>
> [1] GFlowNets and Variational Inference. Malkin et al. ICLR 2023.
>
> [2] PhyloGFN: Phylogenetic inference with generative flow networks. Zhou et al. ICLR 2024

---

> > ### Comment · Reviewer_bMoM · 2024-08-13
> >
> > Thanks for the clear responses. Great paper.

---

### Official Review · Reviewer_c2ea · 2024-07-12

**Soundness:** 3
**Presentation:** 2
**Contribution:** 3
**Rating:** 6
**Confidence:** 4

**Summary:**

The authors provide a framework that allows training of GFlowNet models with streaming data by checkpointing a previously trained GFlowNet model and proposing the streaming balance condition. The problem as well as the approach is well motivated and theoretically sound and this work would be a valuable contribution to the research community. The authors conduct experiments on synthetic tasks and phylogenetic inference, and show improved scalability. Additionally, there is some theory that describes and bounds the approximation error based on errors on the checkpointed GFlowNet model and current estimation problem. Overall, I think the work is quite interesting but it would be nice to consider another experiment since currently most of the experiments are based on synthetic data.

**Strengths:**

- The authors provide useful theoretical contributions that bound the approximation errors into two terms: error from solving the current distribution matching problem and error coming from suboptimal solution to the previous distribution matching problem.
- The problem of training sampling methods that can incorporate streaming data is quite important and relevant to the field, especially for Bayesian Inference methodologies.
- The authors do provide a good set of experiments, and show the benefits of their proposed approach over the standard training of GFlowNets.

**Weaknesses:**

- I found Definition 2 to be quite confusing to read and it is still not clear to me how the equivalence holds. Could the authors clarify what the KL divergence is between, why is it valid and how does it circumvent the problem of parameterizing a partition function?
- The original setup proposed by the authors follows from a trivial extension of the GFlowNets framework which begs to question the novelty of the work.
- It is unclear how their formulation solves the problem of permutation invariance / iid treatment of observations. In particular, given two sets $D_1$ and $D_2$, we know that the posterior distribution $p(x | D_1, D_2)$ can be equivalently written as
$$
p(x | D_1, D_2) = \frac{p(D_1, D_2 | x) p(x)}{p(D_1, D_2)} = \frac{p(D_1 | x) p(D_2 | x) p(x)}{p(D_1, D_2)} = \frac{p(D_1 | x) p(x|D_2)}{p(D_1 | D_2)} = \frac{p(D_2 | x) p(x|D_1)}{p(D_2 | D_1)}
$$
Then, in particular, how is it maintained that given any split of the data $D$ into $D_1$ and $D_2$, the final GFN trained leads to the same solution. If the GFN learns the right solution, then it will follow simply, but it would be nice if the model could satisfy this permutation invariance inherently. Could the authors provide some ablations and control experiments to show to what extent is this constraint satisfied over $t$?
- The authors should add an experiment on discovering causal graphs conditioned on observational samples, which would strengthen their paper considerably.
- The authors should consider the following related work, which are either connected to GFlowNets, sampling from an unnormalized density, or modeling Bayesian posterior inference:

Cranmer, Kyle, Johann Brehmer, and Gilles Louppe. "The frontier of simulation-based inference." Proceedings of the National Academy of Sciences 117.48 (2020): 30055-30062.

Mittal, Sarthak, et al. "Exploring Exchangeable Dataset Amortization for Bayesian Posterior Inference." ICML 2023 Workshop on Structured Probabilistic Inference {\&} Generative Modeling. 2023.

Zhang, Qinsheng, and Yongxin Chen. "Path integral sampler: a stochastic control approach for sampling." arXiv preprint arXiv:2111.15141 (2021).

Richter, Lorenz, et al. "VarGrad: a low-variance gradient estimator for variational inference." Advances in Neural Information Processing Systems 33 (2020): 13481-13492.

Sendera, Marcin, et al. "On diffusion models for amortized inference: Benchmarking and improving stochastic control and sampling." arXiv preprint arXiv:2402.05098 (2024).

Berner, Julius, Lorenz Richter, and Karen Ullrich. "An optimal control perspective on diffusion-based generative modeling." arXiv preprint arXiv:2211.01364 (2022).

Akhound-Sadegh, Tara, et al. "Iterated denoising energy matching for sampling from Boltzmann densities." arXiv preprint arXiv:2402.06121 (2024).

Richter, Lorenz, Julius Berner, and Guan-Horng Liu. "Improved sampling via learned diffusions." arXiv preprint arXiv:2307.01198 (2023).

**Questions:**

- It is not clear what do the authors mean by $\mathbf{A}$ in their notation at the start of Section 2.
- The authors seem to have a typo in the equation under Line 142, essentially the right hand equation should have $p_F^{t+1}$.
- Can the authors provide a visualization of the Set generation task? If I understand correctly, the authors first sample $d$ points randomly and the corresponding $f_i(j)$ randomly as well where $i = 1, ..., K+1$ and $j = 1, ..., d$. Then they define $R_i$ over each set, and a tempered version of $R_i$ with temperature $\alpha$. Is that it?

**Limitations:**

The authors have adequately talked about the limitations and impacts of their work.

---

> ### Author Rebuttal · Authors · 2024-08-03
>
> Thank you for the suggestion of additional experiments and references, which we will include in the revised manuscript. We hope our additional experiments and clarifications enhance your perception of our work. Please let us know if we have left blank spots or if further clarifications are needed.
>
> > I found Definition 2 to be quite confusing to read and it is still not clear to me how the equivalence holds. Could the authors clarify what the KL divergence is between, why is it valid and how does it circumvent the problem of parameterizing a partition function?
>
> Thank you for the opportunity to clarify Definition 2. In fact, there was a typo. The denominator in Eq. 5 should be $p\_{F}^{t}(\tau) f(\mathcal{D}\_{t+1}|x)$ instead of $p\_{F}^{t}(\tau) \pi\_{t+1}(x)$. That being said, Eq. 5 is the KL divergence between the trajectory-level distribution we are learning $p\_{F}^{t + 1}(\tau)$ and that of the previous timestep $p\_{F}^{t}(\tau)$ weighted by the likelihood $f(\mathcal{D}\_{t+1}|x)$ for batch $t+1$. The corrected version of eq. 5 is:
>
> $$
> \mathcal{L}\_{KL}(G\_{t+1} ; G\_{t}) = \mathbb{E}\_{\tau \sim p\_{F}^{(t + 1)}} \left[ \log \frac{p\_{F}^{(t + 1)}(\tau)}{p\_{F}^{(t)}(\tau) f(\mathcal{D}\_{t+1}|x) }\right] \overset{C}{=} \mathcal{D}\_{KL} \left[ p\_{F}^{(t + 1)}(\tau) || p(\tau) \right],
> $$
>
> where $p(\tau) \propto p\_{F}^{t}(\tau) f(\mathcal{D}\_{t+1}|x)$. We hope this correction clarifies definition 2. Otherwise, please let us know.
>
>
> > The original setup proposed by the authors follows from a trivial extension of the GFlowNets framework which begs to question the novelty of the work.
>
> We believe our main contribution is proposing and analyzing the first general-purpose method for streaming approximate Bayesian inference for discrete distributions, including distributions with highly structured supports (e.g., graphs). Additionally, as far as we know, this is the first work explicitly addressing the problem of training GFlowNets in a dynamically evolving environment.
>
> > It is unclear how their formulation solves the problem of permutation invariance / iid treatment of observations
>
> Thanks for the interesting question. As you mentioned, the permutation invariance is clearly satisfied if we perfectly minimize $\mathcal{L}\_{SB}$ (Eq. 4) or $\mathcal{L}\_{KL}$ (Eq. 5) at each time step. However, when this is not the case, the learned distribution is not necessarily invariant to the order in which the datasets $D_{1}, \ldots, D\_{t}$ are observed. To illustrate this, we again examine the problem of phylogenetic inference for two different scenarios highlighted in Fig. 1a of the rebuttal PDF. In both scenarios, we consider that two datasets, $D_{1}$ and $D_{2}$, are observed in sequence and compare the distribution of an SB-GFlowNet trained on the tuples $(D_{1}, D_{2})$ and $(D_{2}, D_{1})$. Fig. 1a (left panel) shows the resulting distributions are not necessarily identical when the GFlowNets at $t = 1$ are not sufficiently well-trained. In contrast, Fig. 1a (right panel) highlights that both distributions are approximately the same when the GFlowNets at $t = 1$ and $t = 2$ are adequately trained. A similar observation holds for the set generation task in Fig. 1b. To the best of our knowledge, this sensibility to the observations’ ordering due to imperfectly approximated posteriors is inherent to any posterior propagation scheme (e.g., streaming VI). We will add this discussion and these experiments to the revised manuscript.
>
> > authors should add an experiment on discovering causal graphs conditioned on observational samples, which would strengthen their paper considerably.
>
> Thank you for the suggestion. We adopted Deleu et al.’s DAG-GFlowNet [1] to address the causal discovery problem with GFlowNets and followed the experimental setup for Fig. 3 of the same paper. Six novel batches of 200 samples were observed --- one at a time for each streaming update. Notably, Fig. 2 of the rebuttal PDF shows SB-GFlowNet accurately matches the target distribution on each streaming update. Similarly, Fig. 3 highlights that the probability of the true DAG characterizing the distribution of the observed data tends to increase along training. We will include these additional experiments in the revised manuscript.
>
> [1] Bayesian Structure Learning with GFlowNets. Deleu et al. UAI 2022
>
> > authors should consider the following related work, which are either connected to GFlowNets, sampling from an unnormalized density, or modeling Bayesian posterior inference
>
> Thank you for the recommendations. We will properly include all related works in the revised manuscript.
>
> > It is not clear what do the authors mean by $𝐴$ in their notation at the start of Section 2.
>
> Indeed, we missed including a definition $A$, which represents the adjacency matrix of the DAG $\mathcal{G}$. We clarify this in the revised manuscript.
>
> > The authors seem to have a typo in the equation under Line 142, essentially the right hand equation should have $p\_{F}^{t + 1}$
>
> Thanks for catching this typo. It should in fact be $p\_{F}^{t +  1}$ instead of $p\_{F}^{t}$. We will update the manuscript accordingly.
>
> > Can the authors provide a visualization of the Set generation task?
>
> We have included an illustration of the set generation task in Fig. 4 of the rebuttal PDF. This task consists of learning to sample from a distribution over sets by iteratively adding elements to an initially empty set. Regarding the definition of each reward $R\_{i}$, it is exactly as you described. Each $f\_{i}(j)$ is independently sampled and fixed before training and, for a set $S$, we let $\log R_{i}(S) = \sum\_{j \in S} f\_{i}(S)$ be the unnormalized log-probability of $S$. The additional presence of $\alpha$ is to evaluate the impact of $R_{i}$’s sparsity on the efficacy of the learning objectives — we repeated the streaming experiment applying different temperatures $\alpha$ to the rewards $R_{i}$ (Table 1), i.e., using $R_{i}^{1/\alpha}$.

---

> > ### Comment · Reviewer_c2ea · 2024-08-12
> > **Reviewer Response**
> >
> > Thanks to the authors for providing a detailed and compelling rebuttal. While most of my concerns have been addressed, I would additionally like to point out that the work would be considerably stronger if the authors could consider a Bayesian linear regression model, with increasing number of observations seen on the X-axis and the KL divergence with the true posterior, which is available in closed form, on the Y-axis.
> >
> > However, I do understand that the discussion period is coming to an end soon and the authors might not have enough time to run this experiment for the discussion. Since all my other concerns have been addressed, I have updated the score accordingly.

---

### Official Review · Reviewer_ZbQa · 2024-07-12

**Soundness:** 3
**Presentation:** 3
**Contribution:** 3
**Rating:** 7
**Confidence:** 4

**Summary:**

This paper introduces Streaming Bayes GFlowNets (SB-GFlowNets), which enables streaming Bayesian inference over discrete parameter spaces, relying on the expressive power of GFlowNets as amortized samplers over discrete compositional objects. The process is akin to Bayesian streaming, where the posterior updates with each new data stream and the posterior from the batch before acts as a prior for the current batch. The authors use a GFlowNet to fit the initial posterior and update it training only on new data stream batches. To do so, the authors propose two different solutions: enforcing a streaming balance condition at each streaming step, or relying on direct divergence-based updates (letting go of the estimation of the normalizing constant). For the latter, the authors show how to obtain low-variance estimators for the gradient of the proposed KL divergence loss. Finally, the authors provide a theoretical performance analysis of their two proposed variants along with a diverse set of experiments to showcase the performance in each case, and show the gain in training time in the streaming setting compared to retraining for all the data at once at each step.

**Strengths:**

- The authors provide a complete study of all the important components within their work (experiments, runtime comparison to full retraining and practical performance enhancement etc).
- The idea is interesting across many applications where data streams are processed continuously.
- The paper is overall well written and easy to follow.

**Weaknesses:**

- I'm not sure where equation (9) comes from. Equating \$\mathcal{L}\_{SB}(\tau)\$ to 0 would result in
\\[
p\_F^{(t+1)}(\tau) = \frac{Z\_t}{Z\_{t+1}} \cdot \frac{p\_B^{(t+1)}(\tau \vert x)}{p\_B^{(t)}(\tau \vert x)} \cdot p\_F^{(t)}(\tau) \cdot f(\mathcal{D}\_{t+1} \vert x)
\\]
Even by assuming \$(p\_F^t, p\_B^t, Z\_t)\$ satisfy TB \$(\mathcal{L}\_{TB} \rightarrow 0)\$, I'm not sure how that would directly yield equation (10). The proofs for proposition 2 also seem to start from (9). Can you elaborate on that?

- To what extent is the assumption that new data is independent of past data under the posterior predictive? This question is especially important in cases where you try to sample from the marginal likelihood over some object (for instance for structure learning), where you lose that independence under the posterior given only the structure (and not parameters for example). I suspect this might be the reason for performance degradation across streaming steps in the Bayesian phylogenetic inference case, given that Felsenstein's algorithm assigns a marginal likelihood.

**Minor Comments**
- Line 30, $\{y_i\}_{i=1}^2$ should be defined/introduced better.
- Line 134, space after comma.
- In line 142, in the second TB condition, it should be $p_F^{(t+1)}(\tau)$ instead of $p_F^{(t)}(\tau)$.
- $p_F^{(t+1)}$ should be evaluated at $\tau$ in equation (10).

**Questions:**

See weaknesses.

**Limitations:**

- The authors adequately addressed limitations throughout the paper (for instance posterior error propagation etc) and through a separate section in the conclusion.

---

> ### Author Rebuttal · Authors · 2024-08-02
>
> Thank you for carefully reviewing our work. We have corrected all the typos you pointed out.
> We hope our clarifications address your concerns and enhance your view of our work.
>
>
> > I'm not sure where equation (9) comes from ...
>
> Thank you for catching this typo, some of the indices were written incorrectly but the result shown in the paper is correct. We have fixed it and added additional clarifications to the paper that we will now explain here.
>
> Indeed, as you’ve written (9) should have $\frac{Z_t}{Z_{t+1}}$ instead of the written $\frac{Z_{t+1}}{Z_{t}}$ and on (10) the left hand side should be $\frac{p_{F}^{t + 1}(\tau)}{p_{B}^{t + 1}(\tau | x)}$.
>
> Then, by assuming that the balance condition is satisfied, we deduce that $Z_t\cdot p_F^t(\tau) = p_B^t(\tau|x)\pi_t(x)$, in other words, $\frac{p_F^t(\tau)}{p_B^t(\tau|x)} = \frac{\pi_t(x)}{Z_t}$.
>
> By doing this substitution on the corrected eq (9), we obtain the corrected eq (10): $$\frac{p_{F}^{t + 1}(\tau)}{p_{B}^{t + 1}(\tau | x)} = Z_{t + 1} \pi_{t}(x) f(\mathcal{D}_{t + 1} | x).$$
>
> Now, as defined in the paper:
>
> $$p\_{\intercal}^{t + 1}(x)= \mathbb{E}\_{\tau \sim p\_{B}^{t + 1}(\cdot | x)}\left[\frac{p\_{F}^{t + 1}(\tau)}{p\_{B}^{t + 1}(\tau | x)}\right],$$ which allows us to conclude, as written in the submitted version, $ p_{\intercal}^{t + 1}(x) \propto \pi_{t + 1}(x)$.
>
> As for the proof of proposition 2, we fix typos and added extra comments to improve the clarity of presentation. Once again we have written $\frac{Z_{t+1}}{Z_{t}}$ instead of the correct $\frac{Z_t}{Z_{t+1}}$. Additionally, this proposition does not use either eq (9) or (10) in the proof. Due to a typo, the equation in lines 532-533 looks similar to eq (9), however, the corrected version, $p\_{\intercal}^{(t + 1), \star}(x) = \frac{Z\_{t + 1}}{Z\_{t}} p\_{\intercal}^{t}(x) f(\mathcal{D}\_{ t + 1 } | x)$ is not related due to the fact that it does not refer to either $p_{F}^{t}(\tau)$ or $p_{F}^{t+1}(\tau)$ but to $p_{\intercal}^{t}(x)$ and $p_{\intercal}^{t+1}(x)$. Once again, we note that these minor corrections do not change the proof significantly and our conclusions still hold.
>
> > To what extent is the assumption that new data is independent of past data under the posterior predictive? This question is especially important in cases where you try to sample from the marginal likelihood over some object (for instance for structure learning), where you lose that independence under the posterior given only the structure (and not parameters for example). I suspect this might be the reason for performance degradation across streaming steps in the Bayesian phylogenetic inference case, given that Felsenstein's algorithm assigns a marginal likelihood.
>
> First, we would like to clarify the assumptions of our model. In the joint distribution, we only require that each data observation is conditionally independent given the parameters, i.e. the variable we want to compute the posterior on. In other words, given a joint distribution:
> $$
> 	p(\theta, \mathcal{D}\_1, \mathcal{D}\_2, \ldots, \mathcal{D}\_4) =
> 	p(\theta) \prod\_{i=1}^4 p(\mathcal{D}\_i | \theta),
> $$
> this only assumes that, given $\theta$, there is independence between the data $\mathcal{D}_i$, however, if we marginalize $\theta$ out of this distribution, the data distribution would be correlated. The fact that the marginal distribution is correlated is not an issue for our method.
>
> In the Bayesian phylogenetics case, the $D_i$ comprises the i-th nucleobase for all biological species. We understand that our description in lines 302-304 might be misleading, since $D_1,\ldots, D_M$ are independent given the tree structure $T$ but  $S_1, \ldots, S_N$ are not independent. We will clarify this in the revised manuscript.
>
> Please let us know if further clarifications are required.

---

> > ### Comment · Reviewer_ZbQa · 2024-08-14
> >
> > Thanks for addressing my comments. I have raised my score to 7.

---

### Official Review · Reviewer_w9BG · 2024-07-13

**Soundness:** 3
**Presentation:** 3
**Contribution:** 4
**Rating:** 6
**Confidence:** 4

**Summary:**

This paper introduces Streaming Bayes GFlowNets, a method for performing approximate Bayesian inference over discrete parameter spaces in streaming data settings.

SB-GFlowNets allow efficient updating of posterior approximations as new data arrives. The method reduces training time compared to retraining GFlowNets from scratch and maintains comparable accuracy.

In conclusion, SB-GFlowNets enable streaming variational inference for discrete parameters, opening up new applications of Bayesian methods to large-scale streaming data problems.

**Strengths:**

- SB-GFlowNets significantly reduce training time compared to retraining GFlowNets from scratch for each new data batch. This is particularly valuable for large-scale streaming data problems.
- The paper provides a theoretical analysis of how errors propagate through posterior updates
- The method is demonstrated to work well on various tasks (set generation, Bayesian linear preference learning, and online phylogenetic inference)

**Weaknesses:**

- As mentioned in the paper's limitations section, inappropriate approximations to earlier posteriors may propagate through time, potentially leading to increasingly inaccurate models. This is a form of catastrophic forgetting
- Due to the error accumulation issue, there may sometimes need to retrain the model from an earlier checkpoint or using the full posterior
- As the paper mentioned, for very sparse target distributions, the performance can degrade, particularly when using the KL-based training scheme

**Questions:**

- Do you think the accumulative error issue can be solved by cache previous gradient (borrowing tricks from continual learning)?
- It would be great if you can just compare with some continual learning method as baselines.
- In sparse target distributions, if not using KL-based training scheme, what are the possible schemes?

**Limitations:**

The authors have adequately addressed the limitations

---

> ### Author Rebuttal · Authors · 2024-08-02
>
> Thank you for thoughtful feedback. We hope our clarifications solve your concerns and elevate your appraisal of our work. Otherwise, we will gladly engage further during the discussion period.
>
> > Do you think the accumulative error issue can be solved by cache previous gradient (borrowing tricks from continual learning)?
>
> Thanks for the question. The accumulative error issue as described in the paper relates to the issue of the past GFlowNets ($t^\prime < t+1$) not being trained until they reach global optima, affecting the quality of the current GFN at time ($t+1$). As we briefly discussed in lines 196-198, our suggestion is to reuse a previous checkpoint at time ($S$)  which has better training loss and use all of the data after that point ($t +1 > t^\prime > S$) to train a GFN for time $t^\prime$ until $t+1$.
>
> The closest parallel to this strategy in continual learning is _replay_. However, replay typically entails using data from past tasks/batches to avoid catastrophic forgetting. We would like to emphasize that there are fundamental differences between catastrophic forgetting and accumulative errors. The most straightforward is that forgetting may happen in continual learning even if we reach local optima consecutively for all data batches. In our case,  we show in the paper when the models converge in our proposed losses, there is no issue with accumulative error.
>
> That being said, including arbitrary terms in our SB-GFlowNets generally entails losing our correctness guarantees. This is the case for replay terms, but also for regularization terms like the elastic weight consolidation. Nonetheless, it is interesting to note both our divergence-based loss in Eq. 5 can be written as
>
> $$
> \mathbb{E}\_{p\_F^{(t+1)}}\left[ - \log \pi\_{t+1}(x) \right] + \mathcal{D}\_{\text{KL}}\left[p\_{F}^{(t+1)} \| p\_{F}^{(t)}\right],
> $$
>
> where the KL term also penalizes forward policies that deviate from that of the previous time step.
>
> We hope this clarification addresses your question. Otherwise, we would be really happy if you could elaborate further on possible connections with continual learning that we could explore during the discussion period.
>
>
> > It would be great if you can just compare with some continual learning method as baselines.
>
> To the best of our knowledge, there is no continual learning method capable of learning discrete posteriors over discrete random variables. If you believe we are missing any relevant piece of work, we would gladly test against them and report our results during the discussion period and add it to the final manuscript.
>
>
> > In sparse target distributions, if not using KL-based training scheme, what are the possible schemes?
>
> As discussed in the paper, the issue with sparse target distributions is that GFlowNets trained with KL-divergence based criteria may fail to represent all modes of the final posterior distribution; we refer to this phenomenon as mode collapse in the paper. An alternative scheme that circumvents this issue (and is another contribution of our work) is to minimize the streaming balance loss (Eq. 4) in an off-policy fashion. In summary, if the target distribution is not sparse, we expect the KL-based training to converge faster, but, on the other hand, streaming balance condition based training can be applied to sparse target distributions.

---

> > ### Comment · Reviewer_w9BG · 2024-08-11
> >
> > Thanks for addressing my question. I am happy to improve my score.

---

### Official Review · Reviewer_xCCn · 2024-07-15

**Soundness:** 4
**Presentation:** 3
**Contribution:** 3
**Rating:** 7
**Confidence:** 3

**Summary:**

The paper introduces a method for Bayesian inference in streaming data scenarios, particularly for discrete parameter spaces, called Streaming Bayes GFlowNets (SB-GFlowNets). The main contributions are:

* Bayesian Streaming Inference: SB-GFlowNets enable the continuous updating of posterior distributions as new data arrives, without needing to recompute from scratch.

* Addressing Intractability: The method addresses the challenge of approximating intractable posteriors in discrete state spaces, which is a limitation of existing variational inference (VI) techniques.

* GFlowNet Utilization: SB-GFlowNets leverage GFlowNets, a class of amortized samplers, to approximate the initial posterior and update it incrementally with new data.

* Case Studies: The effectiveness of SB-GFlowNets is demonstrated in linear preference learning and phylogenetic inference, showing its ability to sample from unnormalized posteriors efficiently in a streaming context.

* Performance: The method is significantly faster compared to repeatedly training a GFlowNet from scratch for the full posterior.

**Strengths:**

Two technically novel training algorithms for streaming inference on GFlowNets based on modified balance condition and VI with a control variate. Presentation was reasonably clear and the experimental results were convincing. Removes limitation of GFLowNets, an increasingly important family of models, to only batched training.

**Weaknesses:**

A weakness of the paper is that it may not sufficiently highlight the significance of streaming inference compared to batch inference. This may be apparent to Bayesian practitioners but not so to a general ML audience. Are there more compelling applications of streaming inference?

**Questions:**

Sentence in lines 58-59 trails off.

**Limitations:**

Yes, limitations have been addressed.

---

> ### Author Rebuttal · Authors · 2024-08-02
>
> Thank you for your suggestions to improve our manuscript. We hope our answers elevate your appraisal of our work. Should you feel that any points require additional clarification, we are more than willing to engage further.
>
> > A weakness of the paper is that it may not sufficiently highlight the significance of streaming inference compared to batch inference. This may be apparent to Bayesian practitioners but not so to a general ML audience. Are there more compelling applications of streaming inference?
>
> Thank you for the suggestion.
>
> One especially compelling application is Bayesian phylogenetics, which we consider in our experiments. Streaming Bayes would allow researchers to update posteriors over the phylogeny tree as they decode new nucleobases in genetic sequences – without reprocessing previously decoded nucleobases. This is particularly convenient since, while there are GPU-accelerated algorithms to compute the likelihood [1-2], “making things fit within a GPU” to minimize communication overhead is a major concern in practical Bayesian phylogenetics. In practice, phylogenetic analyses might involve hundreds of thousands of nucleobases.
>
> More generally, in streaming data settings (i.e., ever-increasing datasets), updating approximate Bayesian posteriors in batch mode entails repeatedly estimating the true posterior from scratch. The major raison d'être for streaming Bayes methods is alleviating this computational bottleneck by reusing the previous posterior estimate as the prior of the next posterior estimate ("Today's posterior is tomorrow's prior').
>
> We will incorporate this discussion in the introduction to make it more suitable for a broader ML audience.
>
>
> > Sentence in lines 58-59 trails off.
>
> Thank you for pointing it out, we accidentally commented out part of this sentence. The intended sentence is as follows: “Notably, despite their successful deployment in solving a wide range of problems ([8 , 9, 19 – 21, 29, 31, 55 ]) , previous approaches assumed that the target (posterior) distribution did not change in time. Hence, this is the first work handling the training of GFlowNets in dynamic environments.”
>
>
>
> [1] Ayres et al., BEAGLE 3: Improved Performance, Scaling, and Usability for a High-Performance Computing Library for Statistical Phylogenetics, Systematic Biology, 2019
>
>
> [2] Suchard, Rambaut. Many-core algorithms for statistical phylogenetics. Bioinformatics, 2009.

---

> > ### Comment · Reviewer_xCCn · 2024-08-13
> > **Response to Rebuttal**
> >
> > I thank the authors for their thoughtful response to my questions and suggestions. That combined with the discussion with other reviewers below, I am glad to increase my score.

---

### Author Rebuttal · Authors · 2024-08-03

Dear reviewers and AC,

We are glad to know reviewers found our work to be valuable contribution to the research community (c2ea), our method to be elegant (bMoM) and our experiment results to be  strong (bMoM), diverse (ZbQa), and convincing (xCCn) — opening a path for new applications of Bayesian methods to large-scale streaming data problem (w9BG).

We are grateful for the reviewers' constructive feedback and suggestions to improve our work. We will incorporate all clarifications requested by the reviewers in the revised manuscript. At the request of reviewer c2ea, we also have run additional experiments in causal discovery.

Best regards,

Authors

---

### Author Response · Authors · 2024-08-14
**Thanks for your service**

Dear reviewers and conference chairs,

We are thankful for your service and commitment to the peer-reviewing process. Also, we greatly appreciate the reviewers’ input, which helped to significantly strengthen and clarify our work. All experiments and discussions will be included in the revised manuscript.

Best regards,

The authors.

---

### Decision · Program_Chairs · 2024-09-25

**Decision:**

Accept (poster)

**Comment:**

This paper considers the setting of streaming Bayesian inference with discrete state spaces. A streaming Bayesian GFlownet (SB-GFlowNet) framework is introduced that uses a GFlowNet and sequentially updates it using new data. Theoretical analysis is provided to understand error propagation in the updates. Empirical studies are performed using SB-GFlowNets for problems involving set generation, linear preference learning, and phylogentic inference.

Overall, reviewers found that the contributions of the paper were novel and that the experimental results were convincing. Several reviewers commented on clarity-related issues, such as notation, definitions, conditions used for the proposition(s), and typos. During the rebuttal phase, these concerns were mostly clarified and addressed, but these revisions still need to be corrected/added in the final manuscript. In particular, after the discussion phase, reviewers unanimously recommended accepting this paper.

Thus, I am happy to recommend acceptance, and strongly encourage the authors to revise the manuscript according to the feedback from reviewers.